# Distinguishing Different Types of Mobile Phone Addiction: Development and Validation of the Mobile Phone Addiction Type Scale (MPATS) in Adolescents and Young Adults

**DOI:** 10.3390/ijerph19052593

**Published:** 2022-02-23

**Authors:** Qing-Qi Liu, Xiao-Pan Xu, Xiu-Juan Yang, Jie Xiong, Yu-Ting Hu

**Affiliations:** 1College of Education for the Future, Beijing Normal University at Zhuhai, Zhuhai 519087, China; liuqingqi@bnu.edu.cn; 2School of Education, Guangzhou University, Guangzhou 510006, China; 3Institute for Public Policy and Social Management Innovation, College of Political Science and Public Administration, Henan Normal University, Xinxiang 453007, China; 4School of Psychology, Central China Normal University, Wuhan 430079, China; yxjccnupsy@126.com; 5Center for Mental Health, Wuhan University of Technology, Wuhan 430063, China; jessiexx@whut.edu.cn; 6School of Business, Jiangnan University, Wuxi 214122, China; hytpsy@yeah.net

**Keywords:** mobile phone addiction type, validity, reliability, adolescents, young adults

## Abstract

Researchers have developed various versions of scales to measure mobile phone addiction. Existing scales, however, focus primarily on the overall level of mobile phone addiction but do not distinguish the potential differences between different types of mobile phone addiction. There is a lack of established scales that can measure different types of mobile phone addiction. The present study aimed to uncover the specific types of mobile phone addiction and develop a Mobile Phone Addiction Type Scale (MPATS) for adolescents and young adults. Adolescents and young adults from two high schools and two universities in Central and South China participated in our study. A total of 108 mobile phone addicts (M_age_ = 17.60 years, SD = 3.568 years; 60.185% males) were interviewed to uncover the specific types of mobile phone addiction. Data from 876 adolescents and young adults (M_age_ = 16.750 years, SD = 3.159 years; 49.087% males) were tested for item discrimination and exploratory factor analysis. Data from 854 adolescents and young adults (M_age_ = 16.750 years, SD = 3.098 years; 50.820% males) were analyzed for construct validity, convergent validity, criterion-related validity, and internal consistency reliability. The 26-item Mobile Phone Addiction Type Scale (MPATS) was developed with four factors named mobile social networking addiction, mobile game addiction, mobile information acquisition addiction, and mobile short-form video addiction. The four-factor, 26-item MPATS revealed good construct validity, convergent validity, criterion-related validity, and internal consistency reliability. The new scale is suitable for measuring different types of mobile phone addiction in adolescents and young adults. Limitations and implications are discussed.

## 1. Introduction

Mobile phone addiction, also known as mobile phone dependence or problematic mobile phone use, is a combined state in which excessive psychological craving and mobile phone overuse lead to significant physiological, psychological, and social impairment [1,2,3]. Adolescents and young adults are an important group of mobile phone users and have a high prevalence of mobile phone addiction [4,5]. Mobile phone addiction can have a severely negative impact on an individual’s physical and mental health, with studies finding that mobile phone addiction can disrupt attention [6]; impair academic performance [7]; reduce life satisfaction [8]; increase depression, anxiety, and stress [9,10]; and potentially lead to sleep disorders, suicidal ideation, and non-suicidal self-injury [11,12,13]. Therefore, it is of great importance to pay in-depth attention to mobile phone addiction in order to promote prevention and intervention practice.

Psychological scales are an indispensable tool for conducting empirical studies on mobile phone addiction. Researchers have developed various versions of scales to measure mobile phone addiction. Some scales are specifically for adolescents, such as the Smartphone Addiction Scale-Short Version for Adolescents [14] and its multiple language versions [15,16,17], while some are specifically for college students, such as the Smartphone Addiction Scale for College Students (Chinese version) [18] and Smartphone Addiction Scale among Medical Students (Malay version) [19]. Some others, however, are applicable to multiple age groups, such as the Mobile Phone Problem Usage Scale [20], Mobile Phone Addiction Index Scale [21], and Problematic Mobile Phone Use Scale [22]. Some scales measure mobile phone addiction through scores on a single dimension [14,15,16,17]. Others, however, contain multiple dimensions that focus on different symptoms of mobile phone addiction and include sub-dimensional scores to reflect specific symptoms of the addiction and the summation of individual dimensional scores to reflect the overall situation of individual mobile phone addiction [18,21]. Nevertheless, there is a lack of established scales that can be used to measure different types of mobile phone addiction.

Because of the diversity of mobile phone functions, different people may have different preferences for mobile phone usage: some are addicted to mobile phone games, some are immersed in mobile social networking services, while others may not be able to control using mobile phones for acquiring various information. Individuals with the same overall level of addiction may have very different types of addictive problems. Previous studies, however, focused primarily on the overall level of mobile phone addiction but did not distinguish the potential differences between different types of mobile phone addiction. Considering only quantitative similarities without looking at detailed qualitative differences in mobile phone addiction may cause undesirable impacts on research findings. Theoretically, distinguishing different types of mobile phone addiction can reveal deep influencing factors and consequences of specific types of mobile phone addiction, which will make the research findings on mobile phone addiction more detailed and more convincing. Practically, distinguishing between different types of mobile phone addiction can promote targeted intervention, which will significantly improve the effectiveness of the intervention. Therefore, it is necessary to distinguish between different types of mobile phone addiction and develop a mobile phone addiction type scale with good reliability and validity so that the focus on mobile phone addiction can be expanded from assessing the levels of severity to also considering the differences in the types of addiction.

Although some research has focused on certain subtypes of mobile phone addiction, such as mobile game addiction [23,24] and mobile social networking addiction [25,26], such studies are relatively few and existing psychological scales on certain subtypes of mobile phone addiction have not been developed and tested for validity and reliability. Compared with Internet addiction, which has been classified into several established types (e.g., Internet game addiction, cyber-relationship addiction, Internet information addiction, and Internet shopping addiction) [27,28,29], it is not known what types of mobile phone addiction are present. Consequently, there is a lack of scales that can be used for measuring different types of mobile phone addiction. Therefore, we conducted interview and questionnaire surveys to uncover the specific types of mobile phone addiction and develop the Mobile Phone Addiction Type Scale (MPATS) among adolescents and young adults. Our study generated an MPATS with good validity and reliability that is suitable for measuring different types of mobile phone addiction in adolescents and young adults.

## 2. Materials and Methods

### 2.1. Participants and Procedure

All procedures and study materials used in the present study were approved by the Ethics Committee at the first author’s affiliation. Informed consent was obtained from all individual participants and the guardians of adolescents under the age of 18. Referring to the common practice of scale development, we set the numbers of subjects of exploratory factor analysis and confirmatory factor analysis at more than 500 in this study. We invited as many participants as possible to improve the representativeness of the samples. Well-trained principal investigators informed the participants that their responses would remain anonymous, that there were no right or wrong answers, and that they could withdraw at any time during the survey. Participants answered the survey in the classrooms during normal school lessons. The data of this study were obtained after excluding the subjects who did not complete the entire questionnaire or provided invalid responses.

The participants in the qualitative study were 108 adolescents from two high schools and two universities in Central and South China, who were nominated by their teachers as having severe mobile phone addiction. They also had scores of 3.5 or higher on a 5-point scale (1–5) on the Mobile Phone Addiction Index scale. Among these participants, 34 were junior high school students, 26 were senior high school students, and 48 were college students; 63 were male and 45 were female. Their ages ranged from 12 to 26 years, with a mean age of 17.60 years (SD = 3.568 years).

The participants of the initial questionnaire that was used for the exploratory factor analysis were 876 students with mobile phone use experience from two high schools and two universities in Central and South China. There were 294 (33.562%) junior high school students, 293 (33.447%) senior high school students, and 289 (32.991%) university students. A total of 430 (49.087%) participants were male and 446 (50.913%) participants were female. Their ages ranged from 12 to 26 years, with a mean age of 16.750 years (SD = 3.159 years).

The participants of the final questionnaire that was used for the confirmatory factor analysis and validity and reliability analysis were 854 students from two high schools and two universities in Central and South China. There were 276 (32.319%) junior high school students, 306 (35.831%) senior high school students, and 272 (31.850%) university students. A total of 434 (50.820%) participants were male and 420 (49.180%) participants were female. Their ages ranged from 12 to 27 years, with a mean age of 16.750 years (SD = 3.098 years).

### 2.2. Materials

#### 2.2.1. Interview Questions

The interview questions included four main parts (see Appendix A, Table A1): basic information, mobile phone usage behaviors, mobile phone addicts’ classification of their mobile phone usage behaviors, and which mobile phone use activities were characterized by behavioral addiction.

#### 2.2.2. Mobile Phone Addiction Index

The Mobile Phone Addiction Index Scale developed by Leung (2008) was used [21]. The scale contains 17 questions measuring four dimensions of mobile phone addiction (i.e., inability to control cravings, anxiety and feeling lost, withdrawal and escape, and productivity loss). The responses are scored on a 5-point scale (1 = never; 5 = always), with higher total scores on the four dimensions indicating approximately severe mobile phone addiction. The scale showed good reliability and validity among both Chinese adolescents and college students [11,30]. In the present study, the alpha coefficient of the scale was 0.851.

#### 2.2.3. Problematic Mobile Phone Use Questionnaire-Short Version

The Chinese version [31] of the Problematic Mobile Phone Use Questionnaire—Short Version [32] was used. The scale contains 11 items measuring a unidimensional factor structure. The responses are scored on a 5-point scale (1 = strongly disagree and 5 = strongly agree), with higher total scores indicating severe problematic mobile phone use. The scale showed good reliability and validity among both Chinese adolescents and college students [31]. In the present study, the alpha coefficient of the scale was 0.938.

#### 2.2.4. Depression Anxiety Stress Scale

The Chinese short version [33] of the Depression Anxiety Stress Scale (DASS) was used [34]. This scale contains a total of 21 questions measuring three dimensions (i.e., depression, anxiety, and stress), with seven questions for each dimension respectively. Participants answered on a 4-point scale of 0–3, with higher scores indicating higher levels of depression, anxiety, and stress. The Chinese short version of the DASS showed good reliability among both Chinese adolescents and college students [35,36]. In this study, the alpha coefficients were 0.898 for the depression dimension, 0.895 for the anxiety dimension, and 0.875 for the stress dimension.

#### 2.2.5. Satisfaction with Life Scale

The Chinese version [37] of the Satisfaction with Life Scale (SWLS) [38] was used. The scale consists of five items and is scored on a 7-point scale (1 = strongly disagree and 7 = strongly agree). The Chinese version of the SWLS showed good reliability and validity among Chinese adolescents and college students and has been widely used in previous research [39,40]. In the present study, the alpha coefficient of the scale was 0.871.

#### 2.2.6. Sleep Disorder Sub-Scale

Sleep quality was assessed using the sleep disorder dimension of the Pittsburgh Sleep Quality Index scale (PSQI) [41], as revised by Liu et al. [42] in Chinese. The scale contains nine questions and is scored on a 4-point scale from 0 to 3, with higher scores indicating more severe sleep disorders. In the present study, the alpha coefficient of the scale was 0.855.

### 2.3. Statistical Analyses

Consensual qualitative research (CQR) [43] was conducted to analyze the qualitative data, on the basis of which, the initial items of the MPATS were compiled. The data from the initial questionnaire were analyzed to explore the factor structure of the scale using the exploratory factor analysis after being tested for item differentiation using the item–total correlation test. According to the results of the item–total correlation test and the exploratory factor analysis, the initial items of the MPATS were censored to generate the revised version of the scale. The data from the final questionnaire were analyzed for construct validity using the confirmatory factor analysis. In addition, the Pearson correlation was conducted to examine the convergent validity and criterion-related validity. The Cronbach’s alpha was analyzed to show the internal consistency reliability.

## 3. Results

### 3.1. Types of Mobile Phone Addiction

Through consensual qualitative research, this study provided a comprehensive classification of the mobile phone use behaviors of adolescent mobile phone addicts and analyzed which of these usage behaviors exhibited the characteristics of behavioral addiction, yielding two main findings, as follows.

The mobile phone use behaviors of mobile phone addicts included eight main types of activities: mobile phone socializing, mobile phone gaming, mobile phone information acquiring, mobile phone short-form video viewing, mobile phone shopping, mobile phone e-book reading, mobile phone music listening, and mobile phone long video viewing.

Only four of the eight mobile phone usage behaviors of mobile phone addicts could be considered fully consistent with the properties of behavioral addiction, namely, mobile social networking behaviors, mobile game behaviors, mobile information acquisition behaviors, and mobile short-form video viewing behaviors. Although all eight mobile phone usage activities are common behaviors among mobile phone addicts, this does not mean that each of these behaviors fully possesses the properties of behavioral addiction, and some mobile phone usage behaviors of mobile phone addicts may also be normal usage behaviors. Strictly speaking, only usage behaviors that fully exhibit the four addictive attributes can be considered mobile phone addiction. While this strict condition is the standard, it is not so strict as to result in normal usage behaviors being classified as mobile phone addiction behaviors. On the other hand, too strict a criterion risks missing certain addictive behaviors. Therefore, this study adopted a middle-ground approach: if a certain usage behavior was not mentioned as having some addictive property in any of the 108 interview cases (e.g., mobile phone e-book reading was not mentioned as being uncontrollable among any of the 108 mobile phone addicts), then and only then would the behavior be considered as not fully possessing the property of addiction, in other words, a very strict exclusion criterion to prevent errors and omissions. An analysis of which mobile phone usage behaviors of mobile phone addicts showed typical characteristics of behavioral addiction revealed that only four mobile phone usage behaviors were represented on all four dimensions of addiction (i.e., inability to control cravings, anxiety and feeling lost, withdrawal and escape, and productivity loss): mobile social networking behaviors, mobile gaming behaviors, mobile information acquiring behaviors, and mobile short-form video viewing behaviors.

Therefore, we classified mobile phone addiction into four types: mobile social networking addiction, mobile game addiction, mobile information acquisition addiction, and mobile short-form video addiction. Given the conceptual content of mobile phone addiction and the typical manifestations of each type of mobile phone addiction mined from the qualitative data, and with reference to specific topics of previous mobile phone addiction scales, we developed the initial version of the MPATS with 32 items (as shown in Appendix A, Table A2). The eight items for each type of mobile phone addiction contained two items for the symptoms of inability to control cravings, two items for the symptoms of anxiety and feeling lost, two items for the symptoms of withdrawal and escape, and two items for the symptoms of productivity loss trait. These items were rated on a five-point scale with 1 indicating “never,” 2 indicating “rarely,” 3 indicating “occasionally,” 4 indicating “often,” and 5 indicating “always.”

### 3.2. Item Discrimination

The item–total correlation test showed that the critical ratio for each item was significant and all were greater than 3 (as shown in Table 1), indicating that the 32 items in the initial version were well differentiated. Therefore, these 32 items were all included in the exploratory factor analysis.

### 3.3. Construct Validity

#### 3.3.1. Exploratory Factor Analysis

The exploratory factor analysis was performed to explore the factor structure of the scale. Prior to the exploratory factor analysis, the data were subjected to a KMO test and Bartlett’s test of sphericity. The KMO value was 0.931 for the 32-item MPATS. The Bartlett’s test of sphericity (*χ*^2^ = 18030.412, *df* = 496, *p* < 0.001) demonstrated that the correlation matrix was suitable for exploratory factor analysis.

The exploratory factor analysis was conducted using principal component analysis with a Promax rotation. Six factors with eigenvalues greater than 1 were extracted, explaining 67.935% of the total variance. Four of the six factors corresponded roughly to the four types of mobile phone addictions, and the other two factors emerged because of the presence of high (above 0.3) double loadings. In order to bring the initial scale more in line with psychometric requirements, items with commonality less than 0.4, factor loading less than 0.4, and the presence of double loadings (both double loadings above 0.3 and the difference between loadings less than 0.3), in consideration of the relationship between the items and the factors, were deleted. Six items were deleted (item 8, item 6, item 31, item 15, item 14, and item 24), leaving 26 items. At this point, a total of four factors were extracted; the first factor (named mobile information acquisition addiction) had seven items, the second factor (named mobile short-form video addiction) had seven items, the third factor (named mobile game addiction) had six items, and the fourth factor (named mobile social networking addiction) had six items. The four factors explained 63.026% of the total variance. The factor loadings and the proportions of explained variance are shown in Table 2.

#### 3.3.2. Confirmatory Factor Analysis

The confirmatory factor analysis was used to test whether the four-factor structure of the scale had good construct validity (see Figure 1). Standardized parameter estimates for all items were statistically significant (*p* < 0.01) and all items had factor loadings above 0.60. The model fit indices were as follows: *χ*^2^/*df* = 4.580, RMSEA = 0.065, CFI = 0.927, NFI = 0.908, and TLI = 0.917. *χ*^2^/*df* was less than 5; RMSEA was less than 0.08; and CFI, NFI, and TLI were all greater than 0.90, indicating that the four-factor structure model fit relatively well and that the scale had good construct validity.

### 3.4. Convergent Validity and Criterion-Related Validity

We tested the correlations between scores of the MPATS and the Problematic Mobile Phone Use Questionnaire—Short Version to analyze whether the MPATS had convergent validity at the between-scale level. As seen in Table 3, the mean scores of different types of mobile phone addiction strongly correlated with scores on the PMPU scale. The correlation coefficients ranged from 0.543 to 0.646, which suggested good convergent validity.

The criterion-related validity of the developed Mobile Phone Addiction Type Scale was examined using depression, anxiety, stress, life satisfaction, and sleep disturbance as indicators. As seen in Table 3, The four types of mobile phone addictions were found to be significantly and positively correlated with depression, anxiety, stress, and sleep disorder, while they were significantly and negatively correlated with life satisfaction. The correlation coefficients of different types of mobile phone addiction and the criterion variables were different, with the strongest correlations occurring between mobile social networking addiction and anxiety, mobile game addiction and depression, mobile information addiction and life satisfaction, and mobile short-form video addiction and sleep disorder. These results indicated that they may have different effects on psychosocial adjustment or be influenced by the different effects of psychosocial adjustment factors. Based on the results of the correlation coefficients, it could be concluded that the MPATS had good criterion-related validity.

### 3.5. Reliability

The results of the reliability analysis are shown in Table 4. The internal consistency coefficient ranged from 0.860 to 0.898, which reflected that all four types of mobile phone addiction exhibited good internal consistency. The correlation coefficients between the four dimensions ranged from 0.501 to 0.599 and did not appear to be too high, indicating that they were both related and distinct and that the separation of the four types was appropriate.

## 4. Discussion

Although previous researchers have developed various versions of mobile phone addiction scales [14,18,19,20,21,22], they all focused on the symptomatic manifestations of mobile phone addiction and lack standardized scales that can reflect the levels of different types of such addiction in individuals. In the present study, the MPATS was developed to have good reliability and validity through a combination of qualitative and quantitative studies, which can properly distinguish between different types of mobile phone addiction and has an important role in promoting research in the field of mobile phone addiction.

In the present study, interviews were conducted with 108 participants who had severe mobile phone addiction. We found four types of mobile phone addiction that existed in these adolescents and young adults: mobile social networking addiction, mobile game addiction, mobile information acquisition addiction, and mobile short-form video addiction. The results were both similar to and different from previous classifications of Internet addiction. In terms of similarities, previous studies on Internet addiction classified Internet addiction into multiple types, such as Internet game addiction, cyber-relationship addiction, Internet information addiction, and Internet shopping addiction [28,29]. The three main forms of Internet addiction, namely, relationship addiction (also known as social addiction), game addiction, and information addiction, were also reflected in the mobile-phone-addicted adolescent and young adult populations. The results suggested a substantial overlap between mobile phone addiction and Internet addiction at the behavioral level, and mobile phone addiction can be considered as the typical manifestation of Internet addiction in the mobile Internet era.

With regard to its differences, compared with the classification of Internet addiction types, this study found that mobile phone short-form video use was also a more dominant mobile phone usage behavior among mobile phone addicts and that such behavior was fully consistent with the four characteristics of behavioral addiction. The results suggested that mobile phone addiction has its unique type of addiction, as opposed to several types of Internet addiction. The Internet environment has changed dramatically in the past decade, where the rapid development of mobile Internet has made certain emerging applications highly characteristic of the mobile network, such as mobile short-form videos. As a three-dimensional medium for information, short-form videos have rich and diverse content and are highly interactive, which can satisfy fragmented entertainment needs and the desire for self-expression in Internet users. According to the China Internet Network Information Center, by June 2021, the number of Chinese short-form video users reached 944 million, accounting for 93.40% of the overall netizen population [44]. Considering that mobile short-form videos use AI technology to develop user preferences and build distribution mechanisms, it is very easy to immerse users in them, leading to overuse (loss of control) [45]. Negative emotions, such as depression and anxiety, may arise when mobile short-form video cannot be used (anxiety and feeling lost). The obvious entertainment value of short-form videos also leads users to use them to escape from and alleviate negative emotions and experiences (withdrawal and escape). The overly fragmented content may also have a negative impact on the cognitive development of individuals, especially on attention control and logical thinking among adolescents and young adults (productivity loss) [46]. Due to these four symptoms, mobile short-form video overuse has become a new type of addiction.

The results of the qualitative study provided a framework for the initial version of the MPATS for adolescents and young adults. Based on the core properties of four types of mobile phone addiction and with reference to the Mobile Phone Addiction Index Scale [21] and the Internet Addiction Type Scale [29], we developed a preliminary scale containing 32 items, of which, eight items were included for each type of mobile phone addiction, and each pair of the eight items reflected one symptom of mobile phone addiction. This ensured that the initial scale took into account the differences in the types of mobile phone addiction without neglecting the symptoms of such addiction. The final results of the exploratory factor analysis yielded 26 questions, of which, four factors were extracted, namely, mobile social networking addiction, mobile game addiction, mobile information acquisition addiction, and mobile short-form video addiction. The four factors explained 63.026% of the total variance, indicating that they could effectively reflect the types of mobile phone addiction among adolescents and young adults.

The results of the reliability analysis showed that the internal consistency coefficients of all four components were high, which suggested that the MPATS had good reliability. Moreover, the correlation coefficients between the four components ranged from 0.501 to 0.599, which are not very high correlations, indicating that the four components were closely related but independent of each other. The confirmatory factor analysis showed that the fit index was good and the scale had good construct validity. In addition, it was found that the correlations between the scores on the Problematic Mobile Phone Use Questionnaire—Short Version and the scores of each type of mobile phone addiction were significant, where the correlation coefficients ranged from 0.543 to 0.646, indicating that the MPATS developed had high convergent validity. By using physical and mental health indicators as correlation indicators, this study also found that all four types of mobile phone addiction were significantly positively correlated with depression, anxiety, stress, and sleep disorders, and significantly negatively correlated with life satisfaction. These findings were relatively consistent with the results of previous studies that did not distinguish between specific types of addiction [2,11,47]. On the other hand, because correlation coefficients are themselves indicators that can be used to compare effect size, comparing correlation coefficients indicated the strongest correlations between mobile social networking addiction and anxiety, mobile gaming addiction and depression, mobile information acquisition addiction and life satisfaction, and mobile short-form video addiction and sleep disturbance. Our results suggested that different types of mobile phone addiction could have different effects on the same psychosocial adjustment indicator or that the same psychosocial indicator could have different effects on different types of mobile phone addiction. These results highlighted the greater need to consider different types of mobile phone addiction as distinct problems.

One of the shortcomings of this study was that a large pool of items containing a large number of questions was not compiled initially because a very limited number of references were available for types of mobile phone addiction. In addition, because of some constraints, expert validation of items was not performed prior to the exploratory factor analysis and confirmatory factor analysis. Although similar issues arose in many studies on scale development, a sound development process is still something that all researchers should try to adhere to. Furthermore, the MPATS treats mobile phone addiction as a continuous variable from low to high and is not applicable to distinguish between mobile phone addiction and non-mobile phone addiction. Currently, popular measurements, such as the Mobile Phone Problematic Use Scale, the Mobile Phone Addiction Index Scale, and the Smartphone Addiction Scale for College Students, all view mobile phone addiction as a continuum from low to high without strictly distinguishing between addiction and non-addiction. However, if a strict distinction between mobile phone addiction and non-addiction can be made, it will greatly help to study the changes in mobile phone addiction propensity from low to high at the quantitative level, as well as grasp the transition of mobile phone addiction from non-addiction to established addiction at the qualitative level.

Despite some shortcomings, the present study is of great significance. Changes in technological content in the Internet era have driven specific changes in the types of technological addiction, and mobile phone addiction in the mobile Internet era has unique attributes that were absent in previously studied Internet addictions. The development of the MPATS is an expansion of the previous Internet Addiction Type Scale and can better reflect the characteristics of the mobile Internet era. It indicates that researchers who are concerned about technological addiction need to pay close attention to the developmental changes in technological content and explore the impact of technology on people’s psyche and behavior on this basis. This study can also provide an instrument with good reliability and validity for researchers to explore and compare different types of mobile phone addiction. Focusing on the specific types of mobile phone addiction can promote the expansion of mobile phone addiction research beyond considering high and low levels of addiction to also considering the differences in types, reveal both quantitative and qualitative changes, and uncover the developmental mechanisms of mobile phone addiction in a more concrete way. Authors should discuss the results and how they can be interpreted from the perspective of previous studies and the working hypotheses. The findings and their implications should be discussed in the broadest context possible. Future research directions may also be highlighted.

## 5. Conclusions

In this study, the qualitative research divided mobile phone addiction into four types: mobile social networking addiction, mobile game addiction, mobile information acquisition addiction, and mobile short-form video addiction. Furthermore, the quantitative research developed a 26-item Mobile Phone Addiction Type Scale (MPATS) to measure these four types of mobile phone addiction. The exploratory factor analysis and confirmatory factor analysis showed that the MPATS had good construct validity. The correlations between the Problematic Mobile Phone Use Scale and four types of mobile phone addiction suggested high convergent validity. The correlations between psychosocial indexes and four types of mobile phone addiction indicated that the MPATS had good criterion-related validity. The internal consistency coefficients reflected high reliability. In conclusion, the MPATS is suitable for measuring different types of mobile phone addiction in adolescents and young adults.

## Figures and Tables

**Figure 1 ijerph-19-02593-f001:**
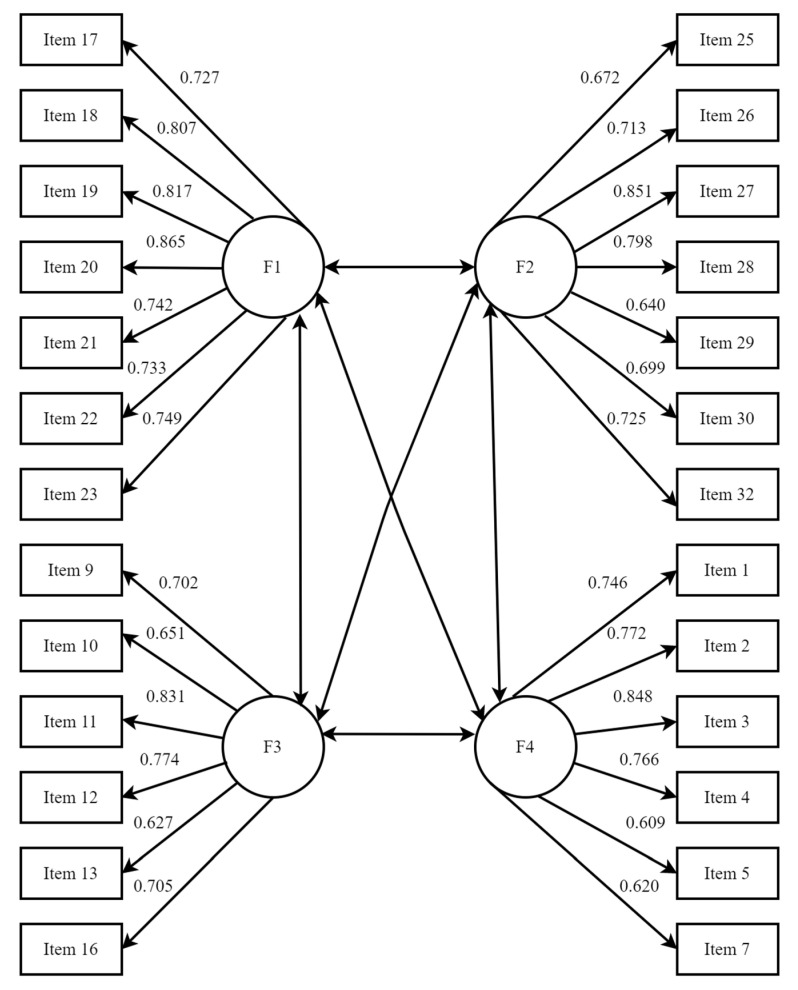
Confirmatory factor analysis of the four-factor model of the Mobile Phone Addiction Type Scale (MPATS).

**Table 1 ijerph-19-02593-t001:** Results of the item discrimination test.

Items	T (Critical Ratio)	*p*
Item 1	20.156	<0.001
Item 2	22.444	<0.001
Item 3	21.318	<0.001
Item 4	20.754	<0.001
Item 5	17.656	<0.001
Item 6	17.892	<0.001
Item 7	22.041	<0.001
Item 8	18.426	<0.001
Item 9	15.456	<0.001
Item 10	16.633	<0.001
Item 11	19.475	<0.001
Item 12	16.510	<0.001
Item 13	17.655	<0.001
Item 14	16.020	<0.001
Item 15	18.066	<0.001
Item 16	16.886	<0.001
Item 17	18.276	<0.001
Item 18	20.748	<0.001
Item 19	17.435	<0.001
Item 20	19.273	<0.001
Item 21	19.027	<0.001
Item 22	20.259	<0.001
Item 23	19.142	<0.001
Item 24	18.962	<0.001
Item 25	19.286	<0.001
Item 26	20.251	<0.001
Item 27	17.550	<0.001
Item 28	17.150	<0.001
Item 29	15.891	<0.001
Item 30	16.791	<0.001
Item 31	19.462	<0.001
Item 32	14.603	<0.001

**Table 2 ijerph-19-02593-t002:** Results of exploratory factor analysis.

Factors	Eigenvalues	Proportion of Explained Variance	Items	Factor Loadings
Factor 1	4.741	18.235%	Item 17. I spend a lot of time searching and browsing for unimportant information on my phone	0.749
			Item 18. I cannot control the amount of time I spend on my phone searching and browsing for information that does not matter.	0.810
			Item 19. I find it especially hard to spend time when I cannot use my phone to search and browse for all kinds of information.	0.676
			Item 20. Even though the information is irrelevant, I still have a hard time controlling myself from searching and browsing on my phone.	0.819
			Item 21. Whenever I get bored, I search or browse for all kinds of information on my phone.	0.808
			Item 22. When I do not know what to do, I keep searching and browsing through all kinds of information on my phone.	0.803
			Item 23. I procrastinate on some things because I am too engrossed in browsing information on my phone.	0.762
Factor 2	4.434	17.053%	Item 25. I can view various short-form mobile phone videos for hours on end and do nothing else.	0.749
			Item 26. I have no concept of time at all when I view short-form videos on my phone.	0.783
			Item 27. I feel particularly uninspired and lost when I cannot view short-form videos on my phone.	0.791
			Item 28. It is hard for me to last long without viewing short-form videos on my phone, even if it is just for a few hours.	0.743
			Item 29. When I am in a bad mood, I watch short-form videos on my phone.	0.753
			Item 30. Because I do not want to face annoying things in my life, I divert my attention by watching short-form videos.	0.754
			Item 32. I spend a lot of time viewing short-form videos on my phone that interfere with my studies and life.	0.618
Factor 3	3.625	13.943%	Item 9. I think the amount of time I spend playing mobile games each day is too short.	0.712
			Item 10. My family or friends complain that I spend too much time playing mobile games.	0.730
			Item 11. I get irritable when I cannot play mobile games for a while.	0.753
			Item 12. When I quit the mobile game, I feel very lost and unhappy.	0.732
			Item 13. When I am depressed, I pick up my phone and play games.	0.666
			Item 16. My relationship with my family has suffered because I am addicted to mobile gaming.	0.666
Factor 4	3.587	13.796%	Item 1. Every chance I get, I open the social networking apps on my phone, even if it is just for a few glances.	0.781
			Item 2. Every day when I wake up, I pick up my phone and swipe through the messages and updates on social media apps.	0.760
			Item 3. I get anxious that I might be missing something when I do not check social apps on my phone for a while.	0.772
			Item 4. I cannot stand not looking at the social apps on my phone for a while.	0.739
			Item 5. When I feel lonely, I interact with people through social apps on my phone.	0.701
			Item 7. I overlook interacting with my family or friends because I spend too much time socializing on my phone.	0.520

**Table 3 ijerph-19-02593-t003:** Results of the convergent validity and criterion-related validity test.

Variables	Mobile Social Networking Addiction	Mobile Game Addiction	Mobile Information Acquisition Addiction	Mobile Short-Form Video Addiction
PMPU	0.646 ***	0.578 ***	0.591 ***	0.543 ***
Depression	0.307 ***	0.424 ***	0.280 ***	0.294 ***
Anxiety	0.416 ***	0.270 ***	0.321 ***	0.317 ***
Stress	0.187 ***	0.260 ***	0.166 ***	0.154 ***
Life satisfaction	−0.358 ***	−0.340 ***	−0.459 ***	−0.369 ***
Sleep disorders	0.324 ***	0.283 ***	0.348 ***	0.449 ***

Note: PMPU—Problematic Mobile Phone Use. *** *p* < 0.001.

**Table 4 ijerph-19-02593-t004:** Internal consistency coefficients and correlations between the four types of mobile phone addiction.

Variables	Internal Consistency	M	SD	1	2	3	4
1. Mobile social networking addiction	0.869	2.380	0.962	1			
2. Mobile game addiction	0.860	2.060	0.914	0.501 ***	1		
3. Mobile information acquisition addiction	0.919	2.057	0.922	0.571 ***	0.557 ***	1	
4. Mobile short-form video addiction	0.898	1.938	0.897	0.548 ***	0.563 ***	0.599 ***	1

Note: *** *p* < 0.001.

## Data Availability

The data presented in this study are available on reasonable request from the corresponding author.

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
