# Peer review of "Distinguishing Different Types of Mobile Phone Addiction: Development and Validation of the Mobile Phone Addiction Type Scale (MPATS) in Adolescents and Young Adults"

_ijerph, 2022, doi:10.3390/ijerph19052593_

Round 1
Reviewer 1 Report
The article presents a scientific solidity in the methodology and results. In this sense, the article has scientific relevance. Authors must improve the theoretical framework with higher quality and variety of bibliography. In the reusltado section, it would be good to include a graphic that makes it more visual. The conclusions should be improved and add logical conclusions from the research and related to other sociological models of addictions to mobiles, technologies or other aspects. Consider the article because of its interest behind the proposed recommendations.Author Response
The article presents a scientific solidity in the methodology and results. In this sense, the article has scientific relevance. Authors must improve the theoretical framework with higher quality and variety of bibliography. In the result section, it would be good to include a graphic that makes it more visual. The conclusions should be improved and add logical conclusions from the research and related to other sociological models of addictions to mobiles, technologies or other aspects. Consider the article because of its interest behind the proposed recommendations.
Response: Thank you very much for your constructive comments. We have improved the Introduction by enhancing the necessity of distinguishing different types of mobile phone addiction in theory and practice. Since there is no theory to distinguish different types of mobile phone addiction, we can not present some specific theoretical evidence on the types of mobile phone addiction in the Introduction section. However, we conducted a qualitative study to classify the mobile phone addiction of adolescents and young adults. The results can be regarded as the theoretical guidance of the types of mobile phone addiction to a certain extent. We included a figure in the Result section to make the results more visual. We also revised the conclusions according to your suggestion.
Reviewer 2 Report
Very interesting article. Question to authors: Is this scale can be applicable in China, or also can be used in other countrys with different life style and internet use traditions?
Author Response
Very interesting article. Question to authors: Is this scale can be applicable in China, or also can be used in other countrys with different life style and internet use traditions?
Response: Thank you very much for highlighting this issue. This scale has good validity and reliability in Chinese adolescents and young adults. Since we have not tested the validity and reliability in populations from other countries, we are unsure whether the scale applies to people in other countries. In the future, we will try to examine the validity and reliability of the MPATS in people from multiple countries.
Reviewer 3 Report
- Abstract
The purpose of the study and the subjects were not described in the abstract. Add information about the purpose of your study and subjects (region, school, gender, age, etc.).
- Introduction
1) Researchers used variables (depression, anxiety, stress, life satisfaction, and sleep disturbance) to determine criterion-related validity. Therefore, please add a review on the relationship between mobile phone addiction and the above variables.
2) As researchers know, there are many tools for smartphone addiction. Please add a little more detail on why development of Mobile Phone Addiction Type Scale (MPATS) is necessary.
- Materials and Methods
1) Ethical issues need to be checked. When conducting research on adolescents, consent must be obtained from the adolescent and his or her guardian to participate in the research. Did you do this?
2) Participants : Describe in detail the subjects. For example, country name, region, school location, gender, etc.
3) There is absolutely no information about how to determine the sample size or data collection procedure. Please describe this in detail.
4) Statistical analysis : Please describe your analysis in detail. For example, what analysis method did you use to identify item differentiation? What did exploratory factor analysis do to confirm? What analyzes were used to test construct validity?
- Results
1) Please unify all subheadings on the same dimension (or level). For example, ‘3.4 Conformal factor analysis’ seems to be better modified with ‘construct validity’. Please correct ‘3.3 Exploratory factor analysis’ and ‘3.6 Reliability analysis’ as above.
2) Please add Figure 1.
3) “Table 3. Results of convergent validity and criterion-related validity analysis” : Since convergent validity and criterion-related validity are not analytical methods, we recommend modifying the ‘analysis’ to ‘test’. Please edit the text content in the same context.
4) “Table 4. Results of Reliability analysis”
Since the values of the correlation coefficients between the four dimensions are presented in table4, it is recommended to revise the table 4 title. And it would be better to present the internal consistency value by adding a space instead of ( ) to improve readability.
- Discussion
1) Uniformity of presentation of statistical values is required (eg 0.501 to 0.599 or 0.54 to 0.65). Please make corrections after checking the guidelines for writing this journal.
- Reference
Make sure all references are listed.
------------------------ The END --------------------
Author Response
#Reviewer 3
- Abstract
The purpose of the study and the subjects were not described in the abstract. Add information about the purpose of your study and subjects (region, school, gender, age, etc.).
Response: Thank you very much for your detailed comments. We have added information about the purpose of the study and the subjects in the abstract.
- Introduction
- Researchers used variables (depression, anxiety, stress, life satisfaction, and sleep disturbance) to determine criterion-related validity. Therefore, please add a review on the relationship between mobile phone addiction and the above variables.
Response: We have added a review on the associations between mobile phone addiction and depression, anxiety, stress, life satisfaction, and sleep disorders.
- As researchers know, there are many tools for smartphone addiction. Please add a little more detail on why development of Mobile Phone Addiction Type Scale (MPATS) is necessary.
Response: According to your suggestion, we have added a little more detail on why development of Mobile Phone Addiction Type Scale (MPATS) is necessary.
- Materials and Methods
- Ethical issues need to be checked. When conducting research on adolescents, consent must be obtained from the adolescent and his or her guardian to participate in the research. Did you do this?
Response: We obtained consent from all adolescents and their guardians.
- Participants: Describe in detail the subjects. For example, country name, region, school location, gender, etc.
Response: We have described the subjects in detail.
- There is absolutely no information about how to determine the sample size or data collection procedure. Please describe this in detail.
Response: We have added some information about how to determine the sample size or data collection procedure.
- Statistical analysis: Please describe your analysis in detail. For example, what analysis method did you use to identify item differentiation? What did exploratory factor analysis do to confirm? What analyzes were used to test construct validity?
Response: According to your suggestion, we have described our analysis in detail.
- Results
- Please unify all subheadings on the same dimension (or level). For example, ‘3.4 Conformal factor analysis’ seems to be better modified with ‘construct validity’. Please correct ‘3.3 Exploratory factor analysis’ and ‘3.6 Reliability analysis’ as above.
Response: We have unified all subheadings on the same dimension.
- Please add Figure 1.
Response: We have added Figure 1.
- “Table 3. Results of convergent validity and criterion-related validity analysis” : Since convergent validity and criterion-related validity are not analytical methods, we recommend modifying the ‘analysis’ to ‘test’. Please edit the text content in the same context.
Response: According to your suggestion, we have modified the word “analysis” to “test”. We also edited the text in the same context.
4) “Table 4. Results of Reliability analysis”
Since the values of the correlation coefficients between the four dimensions are presented in table4, it is recommended to revise the table 4 title. And it would be better to present the internal consistency value by adding a space instead of ( ) to improve readability.
Response: We have revised the title of Table 4. We also presented the internal consistency values by adding a space.
- Discussion
- Uniformity of presentation of statistical values is required (eg 0.501 to 0.599 or 0.54 to 0.65). Please make corrections after checking the guidelines for writing this journal.
Response: We have unified the presentation of statistical values.
- Reference
Make sure all references are listed.
Response: We have checked to make sure all references are listed.
Round 2
Reviewer 3 Report
Thank you for your hard work in editing. However, one part was not corrected. Please correct this part.
- Statistical analysis: Please describe your analysis in detail. For example, what analysis method did you use to identify item differentiation? What did exploratory factor analysis do to confirm? What analyzes were used to test construct validity?
Author Response
Thank you for your hard work in editing. However, one part was not corrected. Please correct this part. Statistical analysis: Please describe your analysis in detail. For example, what analysis method did you use to identify item differentiation? What did exploratory factor analysis do to confirm? What analyzes were used to test construct validity?
Response: Thank you very much for highlighting this issue. Sorry for the misunderstanding; we only revised the statistical analysis in the Results section but did not make a detailed supplement to the Statistical analysis section in the previous manuscript. We have described our analysis in detail: The data from the initial questionnaire were analyzed to explore the construct of the scale using the exploratory factor analysis after being tested for item differentiation using the item-total correlation test. According to the results of the item-total correlation test and the exploratory factor analysis, the initial items of the MPATS were censored to generate the revised version of the scale. The data from the final questionnaire were analyzed for construct validity using the confirmatory factor analysis. In addition, the Pearson correlation was conducted to examine the convergent validity and criterion-related validity. The Cronbach’s alpha was analyzed to show the internal consistency reliability.